# Assessing the Long-Term Role of Vaccination against HPV after Loop Electrosurgical Excision Procedure (LEEP): A Propensity-Score Matched Comparison

**DOI:** 10.3390/vaccines8040717

**Published:** 2020-12-01

**Authors:** Giorgio Bogani, Francesco Raspagliesi, Francesco Sopracordevole, Andrea Ciavattini, Alessandro Ghelardi, Tommaso Simoncini, Marco Petrillo, Francesco Plotti, Salvatore Lopez, Jvan Casarin, Maurizio Serati, Ciro Pinelli, Gaetano Valenti, Alice Bergamini, Barbara Gardella, Andrea Dell’Acqua, Ermelinda Monti, Paolo Vercellini, Giovanni D’ippolito, Lorenzo Aguzzoli, Vincenzo D Mandato, Paola Carunchio, Gabriele Carlifante, Luca Giannella, Cono Scaffa, Francesca Falcone, Stefano Ferla, Chiara Borghi, Antonino Ditto, Mario Malzoni, Andrea Giannini, Maria Giovanna Salerno, Viola Liberale, Biagio Contino, Cristina Donfrancesco, Michele Desiato, Anna Myriam Perrone, Giulia Dondi, Pierandrea De Iaco, Umberto Leone Roberti Maggiore, Mauro Signorelli, Valentina Chiappa, Simone Ferrero, Giuseppe Sarpietro, Maria G Matarazzo, Antonio Cianci, Sara Bocio, Simona Ruisi, Rocco Guerrisi, Claudia Brusadelli, Lavinia Mosca, Raffaele Tinelli, Rosa De Vincenzo, Gian Franco Zannoni, Gabriella Ferrandina, Salvatore Dessole, Roberto Angioli, Stefano Greggi, Arsenio Spinillo, Fabio Ghezzi, Nicola Colacurci, Margherita Fischetti, Annunziata Carlea, Fulvio Zullo, Ludovico Muzii, Giovanni Scambia, Pierluigi Benedetti Panici, Violante Di Donato

**Affiliations:** 1Gynecological Oncology Unit, Fondazione IRCCS Istituto Nazionale dei Tumori di Milano, 20133 Milano, Italy; raspagliesi@istitutotumori.mi.it (F.R.); salvatore.lopez@istitutotumori.mi.it (S.L.); Stefano.ferla1@studenti.unimi.it (S.F.); antonino.ditto@istitutotumori.mi.it (A.D.); umberto.leone@istitutotumori.mi.it (U.L.R.M.); mauro.signorelli@istitutotumori.mi.it (M.S.); valentina.chiappa@istitutotumori.mi.it (V.C.); 2Gynecological Oncology Unit, Centro di Riferimento Oncologico—National Cancer Institute, Via F. Gallini 2, 33081 Aviano, Italy; fsopracordevole@cro.it; 3Woman’s Health Sciences Department, Gynecologic Section, Polytechnic University of Marche, 60126 Ancona, Italy; andrea.ciavattini@ospedaliriuniti.marche.it (A.C.); valentigaetano@gmail.com (G.V.); luca.giannella@ospedaliriuniti.marche.it (L.G.); 4Azienda Usl Toscana Nord-Ovest, UOC Ostetricia e Ginecologia, Ospedale Apuane, 56121 Massa, Italy; alessandro.ghelardi@uslnordovest.toscana.it; 5Department of Clinical and Experimental Medicine, University of Pisa, 56122 Pisa, Italy; tommaso.simoncini@med.unipi.it; 6Gynecologic and Obstetric Unit, Department of Medical, Surgical and Experimental Sciences, University of Sassari, 07100 Sassari, Italy; marco.petrillo@gmail.com (M.P.); dessole@uniss.it (S.D.); 7Gynecologic Oncology Unit, Campus Bio-Medico University of Rome, 00128 Rome, Italy; f.plotti@unicampus.it (F.P.); r.angioli@unicampus.it (R.A.); 8Department of Obstetrics and Gynecology, ‘Filippo Del Ponte’ Hospital, University of Insubria, 21100 Varese, Italy; jvancasarin@gmail.com (J.C.); maurizio.serati@uninsubria.it (M.S.); fabio.ghezzi@uninsubria.it (F.G.); 9Ospedale di Circolo Fondazione Macchi, 21100 Varese, Italy; ciropinelli88@gmail.com (C.P.); rocco@guerrisi.com (R.G.); brusadelli.claudia@gmail.com (C.B.); 10Department of Obstetrics and Gynecology, IRCCS Ospedale San Raffaele, 20100 Milano, Italy; bergamini.alice@hsr.it; 11IRCCS S. Matteo Foundation, Department of Clinical, Surgical, Diagnostic and Paediatric Sciences, University of Pavia, 27100 Pavia, Italy; barbara.gardella@unipv.it (B.G.); arsenio.spinillo@unipv.it (A.S.); 12Gynaecology Unit, Fondazione IRCCS Ca’ Granda Ospedale Maggiore Policlinico, 20100 Milan, Italy; andrea.dellacqua@istitutotumori.mi.it (A.D.); ermelinda.monti@policlinico.mi.it (E.M.); paolo.vercellini@unimi.it (P.V.); 13Division of Obstetrics and Gynecology, Cesare Magati Hospital, Scandiano, Azienda Unità Sanitaria Locale—IRCCS di Reggio Emilia, 42019 Scandiano, Italy; dippolito.giovanni@ausl.re.it (G.D.); aguzzoli.lorenzo@ausl.re.it (L.A.); mandato.vincenzodario@ausl.re.it (V.D.M.); carunchio.paola@ausl.re.it (P.C.); carlinfante.gabriele@ausl.re.it (G.C.); 14Gynecology Oncology Unit, Istituto Nazionale Tumori IRCCS “Fondazione G. Pascale”, 80131 Naples, Italy; c.scaffa@istitutotumori.na.it (C.S.); francesca.falcone@istitutotumori.na.it (F.F.); s.greggi@istitutotumori.na.it (S.G.); 15Department of Obstetrics and Gynecology, S. Anna University Hospital, 44121 Ferrara, Italy; chiara.borghi@unife.it; 16Endoscopica Malzoni, Center for Advanced Endoscopic Gynecological Surgery, 83100 Avellino, Italy; malzonimario@gmail.com; 17Department of Woman’s and Child’s Health, Obstetrics and Gynecological Unit, San Camillo-Forlanini Hospital, 00152 Rome, Italy; andregiannini@tiscali.it (A.G.); salerno.giovannamaria@gmail.com (M.G.S.); 18Department of Obstetrics and Gynecology, Ospedale Maria Vittoria, 10144 Torino, Italy; viola.liberale@gmail.com (V.L.); biagio.contino@aslcittaditorino.it (B.C.); 19Department of Obstetrics and Gynecology, Azienda ASL Frosinone, Ospedale S Trinità di Sora, 03039 Sora, Italy; cristina.donfrancesco@gmail.com (C.D.); micheledesiato@libero.it (M.D.); 20Division of Oncologic Gynecology, Azienda Ospedaliero-Universitaria di Bologna, 40138 Bologna, Italy; myriam.perrone@aosp.bo.it (A.M.P.); giulia.dondi@aosp.bo.it (G.D.); pierandrea.deiaco@unibo.it (P.D.I.); 21Academic Unit of Obstetrics and Gynaecology, IRCCS Ospedale Policlinico San Martino, 16132 Genova, Italy; simone.ferrero@unige.it; 22Department of Neurosciences, Rehabilitation, Ophthalmology, Genetics, Maternal and Child Health (DiNOGMI), University of Genova, 16132 Genova, Italy; 23Department of General Surgery and Medical Surgical Specialties, Gynecological Clinic University of Catania, Via S. Sofia 78, 95124 Catania, Italy; giuseppe.sarpietro@istitutotumori.mi.it (G.S.); mariagraziamatarazzo@gmail.com (M.G.M.); acianci@unict.it (A.C.); 24Department of Gynecology, San Paolo Hospital, Università degli Studi di Milano, 20142 Milan, Italy; sara.bosio@istitutotumori.mi.it (S.B.); simona.ruisi@istitutotumori.mi.it (S.R.); 25Department of Woman, Child and General and Specialized Surgery, University of Campania “Luigi Vanvitelli”, 80138 Naples, Italy; laviniamosca@gmail.com (L.M.); Nicola.colacurci@unicampania.it (N.C.); 26Department of Obstetrics and Gynecology, "Valle d’Itria" Hospital, Martina Franca, Via San Francesco da Paola, 74015 Taranto, Italy; raffaeletinelli@gmail.com; 27UOC Ginecologia Oncologica, Dipartimento per la salute della Donna e del Bambino e della Salute Pubblica, Fondazione Policlinico Universitario A. Gemelli, IRCCS, 00168 Roma, Italy; rosa.devincenzo@unicatt.it (R.D.V.); gianfranco.zannoni@policlinicogemelli.it (G.F.Z.); mariagabriella.ferrandina@policlinicogemelli.it (G.F.); giovanni.scambia@policlinicogemelli.it (G.S.); 28Department of Gynecological, Obstetrical and Urological Sciences, “Sapienza” University of Rome, 00185 Rome, Italy; margherita.fischetti@uniroma1.it (M.F.); ludovico.muzii@uniroma1.it (L.M.); pierluigi.benedettipanici@uniroma1.it (P.B.P.); violante.didonato@gmail.com (V.D.D.); 29Department of Neuroscience, Reproductive Science and Dentistry, School of Medicine, University of Naples Federico II, 80126 Naples, Italy; nunziacarlea@gmail.com (A.C.); fulvio.zullo@unina.it (F.Z.)

**Keywords:** HPV, vaccination, conization, LEEP

## Abstract

*Background*: Primary prevention through vaccination is a prophylactic approach aiming to reduce the risk of developing human papillomavirus (HPV)-related lesions. No mature and long-term data supported the adoption of vaccination in women undergoing conization. *Methods:* This is a retrospective multi-institutional study. Charts of consecutive patients undergoing conization between 2010 and 2014 were collected. All patients included had at least 5 years of follow-up. We compared outcomes of patients undergoing conization plus vaccination and conization alone. A propensity-score matching algorithm was applied in order to reduce allocation biases. The risk of developing recurrence was estimated using Kaplan-Meir and Cox hazard models. *Results*: Overall, charts of 1914 women were analyzed. The study group included 116 (6.1%) and 1798 (93.9%) women undergoing conization plus vaccination and conization alone, respectively. Five-year recurrence rate was 1.7% (*n* = 2) and 5.7% (*n* = 102) after conization plus vaccination and conization alone, respectively (*p* = 0.068). After the application of a propensity-score matching, we selected 100 patients undergoing conization plus vaccination and 200 patients undergoing conization alone. The crude number of recurrences was 2 (2%) and 11 (5.5%) for patients undergoing conization plus vaccination and conization alone, respectively (*p* = 0.231). Vaccination had no impact on persistent lesions (no negative examination between conization and new cervical dysplasia; *p* = 0.603), but reduced the risk of recurrent disease (patients who had at least one negative examination between conization and the diagnosis of recurrent cervical dysplasia; *p* = 0.031). *Conclusions:* Patients having vaccination experience a slightly lower risk of recurrence than women who had not, although not statistically significantly different. Further evidence is needed to assess the cost effectiveness of adopting vaccination in this setting.

## 1. Introduction

Human papillomavirus (HPV) is one of the most common sexually transmitted diseases [1]. HPV is considered the main risk factor for developing cervical cancer and other neoplastic lesions of the lower genital tract and the oropharynx [1]. Currently, several types of HPV have been identified, but only a relatively small number of these—those considered high-risk (HR)—have been recognized as a risk factor for developing pre-neoplastic and neoplastic lesions [2].

Although the prevalence of HPV is high, the proportion of patients with HPV-related lesions and cancer is relatively small. HPV persistence is the main factor influencing the risk of progression into high-grade cervical dysplasia and cancer [2].

In developed countries, the widespread diffusion of secondary prevention reduced the prevalence of invasive cancer. Through the implementation of screening methods (including pap-smear and HPV DNA testing), most pre-neoplastic lesions were detected and treated before the onset of invasive cancer, thus reducing the burden of cervical cancer [3]. Interestingly, the recent adoption of primary prevention (i.e., vaccination) aims to eradicate HPV and HPV-related lesions [4]. Modeling studies showed that the widespread adoption of vaccination would reduce the incidence of HPV related lesions, dramatically [4,5]. With more than 10 years of real-world experience and more than 250 million doses delivered since 2006, HPV vaccines are considered safe and effective. As other prophylactic vaccines, vaccines against HPV would ideally be adopted before having contact with the pathogen of interest. However, there is growing evidence that vaccination against HPV would be useful in women treated for cervical dysplasia. Accumulating data suggested that having vaccination after conization reduced the risk of recurrent cervical dysplasia, basically due to the reduction of new infections. However, data evaluating the role of vaccination after conization are scant and are limited to small comparative series, with short term follow-up [6,7,8,9,10]. Here, we aimed to evaluate the potential role of vaccination in a series of women undergoing conization and 5-year follow-up. As a secondary endpoint, we sought to identify possible factors predicting the risk of cervical dysplasia recurrence among women having the vaccination.

## 2. Materials and Methods

This is a retrospective multi-institutional study conducted in Italy. The Institutional Review Board (IRB) of the Fondazione IRCCS Istituto Nazionale dei Tumori, approved the study (IRB#57/2020). A chart of consecutive patients with newly diagnosed high-grade cervical dysplasia (HSIL) treated in Italy from 1 January 2010 to 31 December 2014 was collected. The present paper is a secondary analysis of research that aimed to identify prognostic factors for developing HSIL recurrence after primary conization and to assess the risk of recurrence based on surgical techniques (laser conization vs. loop electrosurgical excision procedure (LEEP)) [11]. Details of this study are reported elsewhere [11].

The primary endpoint measure of this study was to assess the risk of recurrence comparing women who had conization plus follow-up vs. conization plus vaccination and follow-up. A secondary endpoint measure was to evaluate a possible risk factor for HSIL recurrence in women having HPV vaccination after conization. To avoid possible confounding factors, we just focused on patients undergoing conization with LEEP [11].

The inclusion criteria were: (i) consecutive patients with newly diagnosed HSIL; (ii) the execution of surgical excisional procedure (i.e., conization); (iii) cervical conization performed with LEEP; (iv) conization performed between 2010 and 2014; (v) 5-year follow-up (for non-recurrent patients; while patients developing recurrence within the first 5 years were included even if they did not complete the five years follow-up course). Exclusion criteria were: (i) age < 18 years; (ii) consent withdrawal; (iii) laser conization; (iv) cold knife conization; (v) execution of ablative procedure; (vi) diagnosis of invasive cancer at the time of conization; (vii) glandular lesion; (viii) ongoing pregnancy; and (ix) history of hysterectomy. Importantly, we did not accept the case series of non-consecutive patients. Patients were treated on an outpatient basis using local anesthesia. Procedures were performed under colposcopic guidance, using the LEEP technique [11]. The surgical technique is standardized. Details about surgical treatment are reported elsewhere [11,12,13].

Data were reviewed retrospectively. HPV types were considered high-risk according to the International Agency for Research on Cancer (IARC) [14]. Details of the type of procedures executed, clinical examination, and follow-up were reported elsewhere [11]. Persistent/recurrent cervical dysplasia was defined as the diagnosis of a new HSIL requiring secondary conization or more radical treatments (e.g., hysterectomy). Low-grade cervical lesions (LSIL) were not considered a recurrent disease. Secondary conization for (persistent) LSIL were not considered recurrence when final histological evaluation did not show high-grade lesions. Persistent disease was defined by the diagnosis of HSIL at the first clinical evaluation after primary treatment. Patients having at least one negative examination between primary and secondary treatments were classified as patients with recurrence [11].

### Statistical Methods

Data are summarized using basic descriptive statistics. Since this is a retrospective comparison between two groups (vaccination yes vs. no), we adopted a propensity-matched comparison to avoid the role of possible confounding factors. We developed a multivariable logistic regression model. Age, execution of HPV testing before conization (yes vs. no), the type of HPV involved (HR-HPV yes vs. no or unknown) as well as type of lesion excised (cervical intraepithelial neoplasia (CIN)2 vs. CIN3) were included in the model. A detailed description of the propensity-matched comparison is described elsewhere [15]. Patients who had conization plus vaccination were matched 1:1 to a group of patients who had conization alone. Propensity-score matched analysis attempts to estimate the effect of a treatment by accounting for possible factors, thus minimizing possible allocation biases. Detailed description of statistical methods is reported elsewhere [11]; *p* values < 0.05 were considered statistically significant. GraphPad Prism version 6.0 (GraphPad Software, San Diego, CA, USA) and IBM-Microsoft SPSS version 25.0 (SPSS Statistics. International Business Machines Corporation IBM 2013 Armonk, NY, USA) for Mac were used for the statistical analysis.

## 3. Results

Overall, charts of 2966 patients were retrieved: 567 (19.1%) and 2399 (80.9%) patients undergoing laser conization and LEEP, respectively. Twenty (3.5%) and 155 (6.4%) patients had another conization after laser conization and LEEP, respectively. We restricted our analysis to 2399 patients undergoing LEEP. The median age of the study population was 41 (range 18–89) years. Considering patients with available data (*n* = 1448), HR-HPV was detected in 1371 (94.7%) women.

We focused the analysis on 1914 women with available information regarding the execution of vaccination after conization. Figure 1 shows the flow of patients through the study design. The study group included 116 (6.1%) and 1798 (93.9%) women undergoing conization plus vaccination and conization alone, respectively. Baseline characteristics of the study population are reported in Table 1. The five-year recurrence rate was 1.7% (*n* = 2) and 5.7% (*n* = 102) after conization plus vaccination and conization alone, respectively (*p* = 0.068). Figure 2A shows the 5-year recurrence rate according to vaccination status.

Since patients’ characteristics between the two groups were not fully comparable, basically due to the retrospective observational nature of the study, we applied a propensity-score matching comparison. Table 2 reports the baseline characteristics of the two study groups. Overall, 86 out of 100 women undergoing conization plus vaccination had details regarding vaccination (doses administered and timing). All those women (*n* = 86, 100%) had at least two doses of vaccines; three doses were completed in 68 (79%) women. Looking at the timing of vaccination, 70 (81.4%) and 12 (14%) women had the first dose within the first month and the first three months after conization, respectively. Four (4.6%) women had vaccination between 3 and 6 months after conization.

Overall, 7% and 93% of patients had bivalent and quadrivalent vaccination, respectively. The crude number of persistence/recurrences was 2 (2%) and 11 (5.5%) for patients undergoing conization plus vaccination and conization alone, respectively (*p* = 0.231). Recurrence-free survival is reported in Figure 2B. After the adjustment for margin status, we observed vaccination was more likely to be useful in women with negative margins in comparison to women with positive margins (*p* = 0.06, log-rank test; Figure 3A). After the adjustment for HPV persistence, we observed vaccination was more likely to be useful in women with HPV persistence in comparison to women without viral persistence (*p* = 0.06, log-rank test; Figure 3B). Appendix A reports univariate and multivariate analysis evaluating factors predicting persistence/recurrence. At univariate analysis, positive margins (especially endocervical ones) and HPV persistence were associated with an increased risk of recurrence. However, via multivariate analysis, no factor was associated with recurrence. No case of invasive cancer was observed at the time of recurrence and all women were diagnosed with HSIL. The two patients developing persistence/recurrence after primary conization and vaccination (both performed within the first month) and two patients developing recurrence after conization alone were classified as persistence of the lesions (*p* = 0.603). All of them had positive surgical margins and persistence of HPV infection. Recurrent disease (including patients who had at least one negative examination between conization and the diagnosis of recurrent HSIL) was observed in 0 and 9 (4.5%) patients in the conization plus vaccination and conization alone groups, respectively (*p* = 0.031, Fisher exact test). In our series, timing of vaccination (first dose < 1 month (2/70) vs. 1–3 months (0/12) vs. >3 months (0/4)) did not correlate with the risk of recurrence (*p* = 1.00).

## 4. Discussion

The present study investigated the role of vaccination against HPV after conization in reducing recurrent HSIL. This is a retrospective multi-institutional study including only patients with at least five years of follow-up. This study reports several noteworthy findings. First, we observed that vaccination is associated with a slightly non-significant reduction in the recurrence rate in the whole population. Second, after the correction of various confounding factors, vaccination resulted in the only modifiable factor that might slightly reduce the recurrence rate. Third, we observed that the benefit of vaccination is more evident in patients with negative margins (at low risk of persistence of the lesions).

Accumulating data supported the adoption of vaccination after conization, since it is the only modifiable factor that might reduce the recurrence rate [6,7,8,9,10,16]. In 2013, Kang et al. evaluated the risk of recurrence after LEEP in women having or not having vaccination [6]. The study included 360 and 377 women who had conization plus vaccination and conization alone, respectively. The recurrence rate was 2.5% and 7.2% following conization plus vaccination and conization alone, respectively [6]. The omission of vaccination after conization was an independent risk factor for cervical dysplasia recurrence (*p* < 0.001) [6]. The SperAnZA study (Sperimentazione anti-HPV zona Apuana) is a prospective (non-randomized) study comparing outcomes of women undergoing conization plus vaccination vs. conization alone [7]. The study showed the clinical effectiveness of 80% in reducing the recurrence rate. The study included 398 patients undergoing conization with at least six months follow-up. The median follow-up of the study population was 27 months. The recurrence rate was 1% and 6% following conization plus vaccination and conization alone, respectively [7]. Petrillo et al. reported outcomes of a study including 182 and 103 women undergoing conization plus vaccination vs. conization alone, respectively. Recurrence rate was 2.3-fold higher for women undergoing conization alone in comparison to women undergoing conization plus vaccination [8]. Our findings corroborate the results of these studies. However, the present study is the only investigation assessing 5-year outcomes.

One of the most interesting features regarding vaccination is the timing of vaccination. Growing evidence supports that the early administration of vaccination improves the efficacy of the protection against HPV [8]. Sand et al. reported data of 17,128 women (including 2074 women who had vaccination). There was a statistically non-significant lower risk of cervical dysplasia recurrence among vaccination (HR: 0.86, 95% CI: 0.67–1.09). Women vaccinated 0–3 months before conization experience a slightly lower risk of recurrence (HR adjusted: 0.77, 95% CI: 0.45–1.32) than women vaccinated 0–12 months after conization (HR adjusted: 0.88, 95% CI: 0.67–1.14), although not statistically significantly different [9]. Interestingly, in two post hoc analyses of the FUTURE II and the PATRICIA trials, women undergoing treatment for cervical intraepithelial neoplasia after vaccination had a reduced risk of subsequently developing recurrent cervical dysplasia [17,18]. However, other studies did not support the adoption of vaccination in women with cervical dysplasia [10].

The inherent biases related to the retrospective study design represent the main weaknesses of the present study. Moreover, the present study had several limitations: (i) the lack of data of several important patients’ constitutional variables (including smoking history); (ii) the selective reporting bias; (iii) the small sample size of the vaccinated cohort (however, a post-hoc analysis suggested that about 800 patients per arm are necessary to demonstrate a statistically significant benefit of the incorporation of vaccination in women undergoing conization); (iv) our population included only women younger than 45 years of age, thus our results are not projectable in all clusters of age (however, this population represents the ideal target for receiving vaccination); (v) we have no data on the possible execution of prophylactic vaccination before conization; (vi) owing to the small size, we cannot test the impact of number of doses administered and timing of first dose administered on the efficacy of the vaccine. These features might impact the interpretation of our results. The main strength of the study is the analysis of a large database including consecutive patients undergoing conization with long-term follow-up. Additionally, this investigation reflects the Italian situation during the years 2010–2014. In particular, it is interesting to note that only 6% of women receiving conization had a vaccination, thus highlighting the low proportion of women receiving vaccination after cervical conization during the years of the study. However, in Italy, guidelines only started to recommend vaccination in women undergoing conization in 2020 [19]. One of the main interesting points deserving attention is the timing of vaccination. Accumulating evidence underlines that having the vaccination before conization might improve the outcomes of the patients [9]. Additionally, a late vaccination might fail to prevent re-infection in women at risk. Further evidence is necessary to identify the cohort of women who can have much more benefit from vaccination, thus reducing the burden for the healthcare.

## 5. Conclusions

The present paper evaluated the impact of vaccination among women undergoing conization due to high-risk cervical dysplasia. Our study highlighted that the adoption of vaccination slightly reduces the recurrence rate. Vaccination is one modifiable factor that might improve the outcomes of women affected by high-grade cervical dysplasia. Further evidence is warranted to assess the cost-effectiveness of the adoption of vaccination in this cluster of patients. Additionally, those findings would be useful to counsel patients about their risk of recurrence, and to tailor surveillance based on various risk and protective factors.

## Figures and Tables

**Figure 1 vaccines-08-00717-f001:**
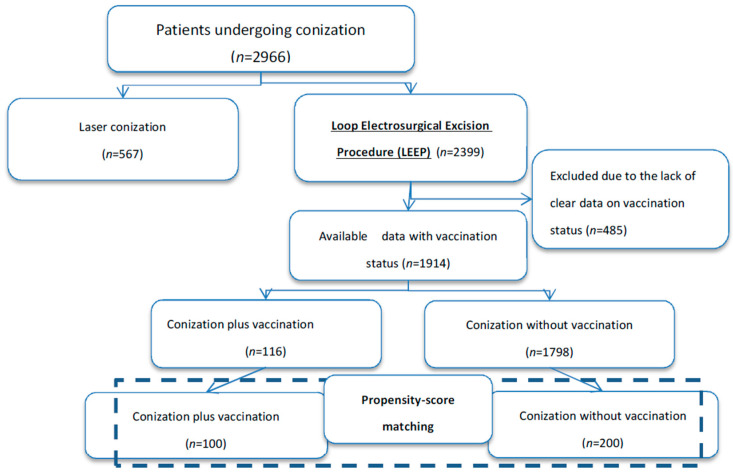
Study design.

**Figure 2 vaccines-08-00717-f002:**
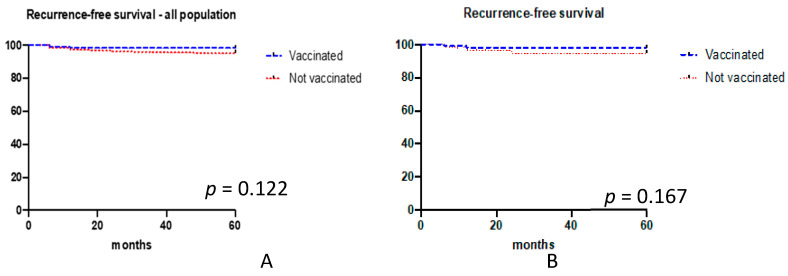
Five-year recurrence for women undergoing conization plus vaccination vs. conization alone (**A**) and five-year recurrence in the propensity-matched comparison cohorts (**B**).

**Figure 3 vaccines-08-00717-f003:**
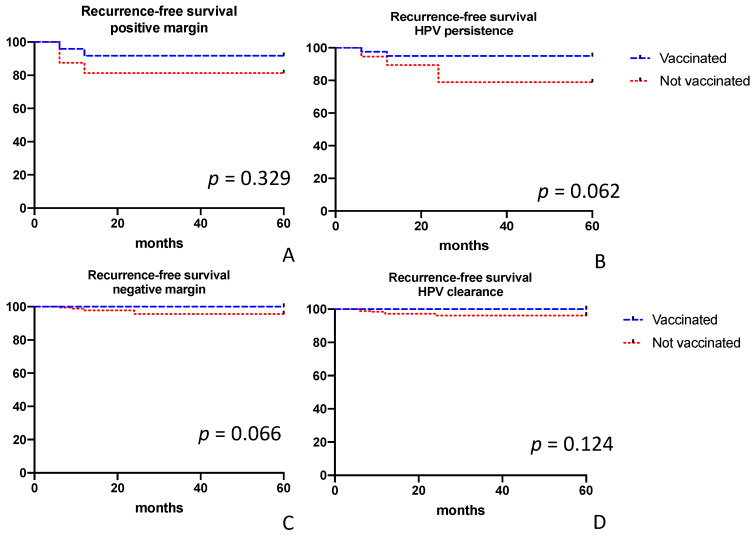
Recurrence rate according to various risk factors: positive margins (**A**), HPV persistence (**B**), negative margins (**C**), and HPV clearance (**D**).

**Table 1 vaccines-08-00717-t001:** Baseline characteristics of the population.

Characteristics	Whole Study Population (*n* = 1914)	Conization Plus Vaccination (*n* = 116)	Conization Alone (*n* = 1798)	*p* Value (Vaccinated vs. Not Vaccinated)
Age, years	39 (17–89)	35 (24–45)	39 (17–89)	<0.001
Body mass index	22.85 (14.4–44)	21 (17–33)	23 (14.4–44)	<0.001
Menopause				
No	1545 (80.7%)	116 (100%)	1429 (79.5%)	<0.001
Yes	369 (19.3%)	//	369 (20.5%)	
Reason for conization				0.066
CIN2	827 (43.2%)	60 (51.7%)	767 (42.7%)	
CIN3	1087 (56.8%)	56 (48.3%)	1031 (57.3%)	
High-risk HPV involved *				<0.001
No	1026 (53.6%)	41 (35.3%)	985 (54.8%)	
Yes	888 (46.4%)	75 (64.7%)	813 (45.2%)	
Positive margins	189 (9.9%)	25 (21.5%)	164 (9.1%)	<0.001
Endocervical	135 (7%)	19 (16.4%)	116 (6.4%)	<0.001
Esocervical	55 (2.9%)	6 (5.2%)	49 (2.7%)	0.142
HPV persistence **				
No	1053 (55%)	60 (51.7%)	993 (55.2%)	0.501
Yes	335 (17.5%)	51 (44%)	284 (15.8%)	<0.001
Unknown	526 (27.5%)	5 (4.3%)	521(29%)	<0.001
Recurrence	104 (5.4%)	2 (1.7%)	102 (5.7%)	0.068

Data are reported as number (%) and median (range); Abbreviations: CIN, cervical intraepithelial neoplasia; HPV, human papillomavirus; *, data on HPV involved in HSIL/CIN2+ were calculated on the basis of 1597 patients undergoing HPV testing before conization; **, data on HPV persistence were calculated on 1516 patients undergoing HPV testing after conization.

**Table 2 vaccines-08-00717-t002:** Baseline characteristics of the population included in the propensity score matching.

Characteristics	Conization Plus Vaccination (*n* = 100)	Conization Alone (*n* = 200)	*p* Value
Age, years	33.5 (24–43)	33.3 (24–44)	0.895
BMI, kg/mq	21 (17–33)	21.1 (17–32.8)	0.867
Menopause			1.00
No	100 (100%)	200 (100%)	
Yes	0	0	
HR-HPV detected			0.895
No/Unknown	31 (31%)	64 (32%)	
Yes	69 (69%)	136 (68%)	
Not tested			
Type of cervical dysplasia			0.902
CIN2	54 (54%)	106 (53%)	
CIN3	46 (46%)	94 (47%)	
Positive margins			1.00
No	76 (76%)	151 (75.5%)	
Yes	24 (24%)	49 (24.5%) **	
Type of involved margins			
Endocervical	18 (18%)	37 (18.5%)	1.00
Esocervical	6 (6%)	13 (6.5%)	1.00
HPV persistence *			0.707
No/Unknown	62 (62%)	118 (59%)	
Yes	38 (38%)	82 (41%)	
Recurrence/Persistence			
No	98 (98%)	189 (94.5%)	0.231 (yes vs. no)
Yes, persistence	2 (2%)	2 (1%)	0.603 (persistence vs no)
Yes, recurrence	0	9 (4.5%)	0.031 (recurrence vs. no)

Data are reported as median (range) and number (%); Abbreviation: BMI, body mass index; CIN, cervical intraepithelial neoplasia; HR, high-risk; HPV, human Papillomavirus; *, data on HPV persistence were calculated only for patients undergoing HPV testing after conization; **, one patient had both esocervical and endocervical positive margins.

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
