# Peer review of "Assessing the Long-Term Role of Vaccination against HPV after Loop Electrosurgical Excision Procedure (LEEP): A Propensity-Score Matched Comparison"

_vaccines, 2020, doi:10.3390/vaccines8040717_

Round 1

Reviewer 1 Report

In this manuscript, Bogani et al. evaluated the impact of HPV vaccination among women undergoing conization due to high-risk cervical dysplasia. Whilst the manuscript offers novel insights, the presentation of the dat should be improved before publication.

  • All of the graphs should be uniform and the same size - particularly, the graphs in figure 2 and 3 should be larger, in line with figure 4.
  • For figure 4, the graphs should be labelled a-d and the legend should be extended in line with this.
  • p values should be written as p=X.XXX, not p=X,XXX.

Author Response

In this manuscript, Bogani et al. evaluated the impact of HPV vaccination among women undergoing conization due to high-risk cervical dysplasia. Whilst the manuscript offers novel insights, the presentation of the data should be improved before publication.

Comment 1: All of the graphs should be uniform and the same size - particularly, the graphs in Figures 2 and 3 should be larger, in line with figure 4.

Answer: In order to comply with the reviewer’s comment we modified the figures, accordingly.

Comment 2: For figure 4, the graphs should be labeled a-d and the legend should be extended in line with this.

Answer: In order to comply with the reviewer’s comment we modified the figures, accordingly. Please, note that according to the request of reviewer#3 we condensed Figure 2 and Figure 3 into a single Figure (new Figure 2); then Figure 4 now is called Figure 3

Comment 3: p values should be written as p=X.XXX, not p=X,XXX.

Answer: In order to comply with the reviewer’s comment we modified the p values, accordingly.

Reviewer 2 Report

The authors presented here an interesting article about the vaccination against HPV after LEEP, which could provide important reference for clinical treatment.

I have several questions:

1) What the percentage of LEEP treatment since the authors excluded other surgeries such as laser and cold knife? What is the recurrence of other surgery treatment? This will affect the importance of this paper.

2)Since the authors suggested that vaccination at the early time might help prevent recurrence better, it might be better to subgroup patients with vaccination based on different time scale and then compare them.

3)It's no clear to me if vaccination more than once would reduce the recurrence or not. 

Author Response

The authors presented here an interesting article about the vaccination against HPV after LEEP, which could provide an important reference for clinical treatment.

I have several questions:

Comment 1: What the percentage of LEEP treatment since the authors excluded other surgeries such as laser and cold knife? What is the recurrence of other surgery treatment? This will affect the importance of this paper.

Answer: We thank the reviewer for this comment. In our series, more than 80% of patients had LEEP. We decided to include only patients undergoing LEEP to reduce possible confounding factors. In order to comply with the reviewer’s comment, we added the required data at the beginning of the results section.

Comment 2) Since the authors suggested that vaccination at an early time might help prevent recurrence better, it might be better to subgroup patients with vaccination based on the different time scales and then compare them.

Answer: It is a very interesting point too. In our series, the timing of vaccination did not correlated with the risk of recurrence. Two, 0 and 0 recurrences were observed, among  patients having first dose <1month (n=70), between 1 and 3 months (n=12), and >3 months (n=4). Additionally, we added a comment in the discussion regarding this finding. The small sample size did not allow us to assess the clear impact of having an early or late vaccination.  

Comment 3) It's no clear to me if vaccination more than once would reduce the recurrence or not. 

Answer: We thank the reviewer for the present comment, that is very interesting. To date, there are no data to clarify this point. In order to comply with the reviewer’s comment, we addressed this point in the discussion. 

Reviewer 3 Report

The manuscript illustrates the results of a retrospective multicenter study on the impact of HPV vaccination after surgical LEEP treatment of high-grade cervical lesions. The topic is of interest but the study period refers to years when this practice was rarely performed. As a consequence, the number of women undergoing conization plus vaccination (N=116) is much smaller than the number of women undergoing conization without vaccination (N=1798). To overcome this imbalance, the authors performed a propensity-score matching (1:2) of women with (N=100) and without (N=200) vaccination, and these two groups are well balanced for several characteristics (described in Table 4). 

The five-years recurrence rate was lower among vaccinated than among unvaccinated women, but the authors do not correctly distinguish between persistent (i.e., detected at first follow-up) and recurrent (i.e., detected after one or more negative follow-up visit) lesions.

Specific comments:

-Introduction, line 92: "HPV is considered the main risk factor for developing......and the head and neck district" should be corrected in "the oropharynx".

-Materials and Methods: the study population included in the present study is the same described in a recently published paper by the same authors (Gynecol Oncol, https://doi.org/10.1016/j.ygyno.2020.08.025), that the authors omitted to cite. Please, add this reference.

-Results, lines 220-224, and Table 4: the 2 cases of high-grade lesion developed after conization plus vaccination are described as persistence in the text and recurrence in the table; please, verify and correct accordingly. This is an important point because thevaccination  impact analysis performed separately for persistent and recurrent lesions would figure 0% vs 5.5% recurrent lesions in vaccinated and unvaccinated women, respectively, possibly reaching statistical significance.

-Discussion, lines 281-282: the sentence would better read as "Our findings corroborate the results of these studies."

-Discussion, lines 313-314: the sentence on late vaccination is not clear; it probably implies that a late vaccination might "fail to prevent" rather than "increase" re-infection rate.

-Conclusions, lines 318 and 320: "high-risk" should be substituted by "high-grade" since it refers to the lesions and not to the HPV types.

-Figures: the figure of page 7 has no title; according to the text, it should be Figure 4a, but it doesn't correspond. I would suggest to combine the two figures in page 6 as Figure 2, A and B, and name the figure in page 6 as Figure 3, A and B. Figure 4 has two panels, that need to be indicated in the figure and the title.

Author Response

The manuscript illustrates the results of a retrospective multicenter study on the impact of HPV vaccination after surgical LEEP treatment of high-grade cervical lesions. The topic is of interest but the study period refers to years when this practice was rarely performed. As a consequence, the number of women undergoing conization plus vaccination (N=116) is much smaller than the number of women undergoing conization without vaccination (N=1798). To overcome this imbalance, the authors performed a propensity-score matching (1:2) of women with (N=100) and without (N=200) vaccination, and these two groups are well balanced for several characteristics (described in Table 4). 

Comment 1: The five-year recurrence rate was lower among vaccinated than among unvaccinated women, but the authors do not correctly distinguish between persistent (i.e., detected at first follow-up) and recurrent (i.e., detected after one or more negative follow-up visits) lesions.

Answer: We thank the reviewer for the comment, we agree with the reviewer's comment. We clarified this point in the text and in Tables (as reported below).

Additionally, we would like to thank the reviewer' for all these comments, his/her comments were very useful in improving the quality of the paper. 

Specific comments:

Comment 2) Introduction, line 92: "HPV is considered the main risk factor for developing......and the head and neck district" should be corrected in "the oropharynx".

Answer: In order to comply with the reviewer’s comment, we modified the text, accordingly.

Comment 3) Materials and Methods: the study population included in the present study is the same described in a recently published paper by the same authors (Gynecol Oncol, https://doi.org/10.1016/j.ygyno.2020.08.025), that the authors omitted to cite. Please, add this reference.

Answer: In order to comply with the reviewer’s comment, we added the citation of our previous paper.

“The present paper is a secondary analysis of a research aimed to identify prognostic factors for developing HSIL recurrence after primary conization and to assess the risk of recurrence based on surgical techniques (laser conization vs. loop electrosurgical excision procedure (LEEP)) [11]. Details of this study are reported elsewhere [11]. ”

11- Bogani G, DI Donato V, Sopracordevole F, et al. Recurrence rate after loop electrosurgical excision procedure (LEEP) and laser Conization: A 5-year follow-up study. Gynecol Oncol. 2020 Sep 3:S0090-8258(20)33826-9. doi: 10.1016/j.ygyno.2020.08.025. Epub ahead of print. PMID: 32893030.

Comment 4) Results, lines 220-224, and Table 4: the 2 cases of high-grade lesion developed after conization plus vaccination are described as persistence in the text and recurrence in the table; please, verify and correct accordingly. This is an important point because the vaccination impact analysis performed separately for persistent and recurrent lesions would figure 0% vs 5.5% recurrent lesions in vaccinated and unvaccinated women, respectively, possibly reaching statistical significance.

Answer: We thank the reviewer for this observation. The two patients had persistent disease. We clarified this point and corrected the data, accordingly.

Comment 5) Discussion, lines 281-282: the sentence would better read as "Our findings corroborate the results of these studies."

Answer: In order to comply with the reviewer’s comment, we modified the text, accordingly.

Comment 6) Discussion, lines 313-314: the sentence on late vaccination is not clear; it probably implies that a late vaccination might "fail to prevent" rather than "increase" re-infection rate.

Answer:  Yes, we agree with the reviewer’s comment. We thank the reviewer for this observation. In order to comply with the reviewer’s comment, we modified the text, accordingly.

Comment 7) Conclusions, lines 318 and 320: "high-risk" should be substituted by "high-grade" since it refers to the lesions and not to the HPV types.

Answer:  Yes, we agree with the reviewer’s comment. We thank the reviewer for this observation. In order to comply with the reviewer’s comment, we modified the text, accordingly.

Comment 8) Figures: the figure of page 7 has no title; according to the text, it should be Figure 4a, but it doesn't correspond. I would suggest to combine the two figures in page 6 as Figure 2, A and B, and name the figure in page 6 as Figure 3, A and B. Figure 4 has two panels, that need to be indicated in the figure and the title.

Answer:  In order to comply with the reviewer’s comment, we modified the figures, accordingly.

Thank you 

Round 2

Reviewer 2 Report

I appreciate the authors spent the effort and time to answer my questions and I'm satisfied with the answers. In figure 1, the lines (box line) of the right part are missing, please change it.

Author Response

Comment: I appreciate the authors spent the effort and time to answer my questions and I'm satisfied with the answers. In figure 1, the lines (box line) of the right part are missing, please change it.

Answer: We thank the reviewer for his/her efforts in helping us to improve the manuscript. In order to comply with the reviewer's comment, we modified the figure